



# The Microfluidic Ice Nuclei Counter Zürich (MINCZ): A platform for homogeneous and heterogeneous ice nucleation

Florin N. Isenrich[1*], Nadia Shardt[2*], Michael Rösch[2], Julia Nette[1], Stavros Stavrakis[1], Claudia Marcolli[2], Zamin A. Kanji[2], Andrew J. deMello[1], Ulrike Lohmann[2]

[1]Institute for Chemical and Bioengineering, ETH Zurich, Zürich, 8093, Switzerland
[2]Institute for Atmospheric and Climate Science, ETH Zurich, Zürich, 8092, Switzerland

*These authors contributed equally to this work.

**Correspondence**: Nadia Shardt (nadia.shardt@env.ethz.ch) and Andrew J. deMello (andrew.demello@chem.ethz.ch)

**Abstract.** Ice nucleation in the atmosphere is the precursor to important processes that determine cloud properties and lifetime. Computational models that are used to predict weather and project future climate changes require parameterizations of both homogeneous nucleation (i.e., in pure water) and heterogeneous nucleation (i.e., catalysed by ice-nucleating particles, INPs). Microfluidic systems have gained momentum as a tool for obtaining such parameterizations and gaining insight into the stochastic and deterministic contributions to ice nucleation. In this regard, polydimethylsiloxane (PDMS) devices are typically used to generate droplets in microchannels that are then cooled and monitored "on-chip". However, using PDMS has two drawbacks. First, it has a low thermal conductivity that generates temperature gradients within a PDMS chip upon cooling from below, which can lead to increased temperature uncertainty at the droplets' location. Second, it readily absorbs water and is gas permeable, which compromises the stability of droplets over extended timescales. To overcome these shortcomings, we have developed a new instrument: the Microfluidic Ice Nuclei Counter Zürich (MINCZ). In MINCZ, droplets are generated using a PDMS chip, but are then stored in fluoropolymer tubing that is relatively impermeable to water and solvents. Droplets within the tubing are cooled in an ethanol bath that ensures efficient heat transfer and reduces uncertainty in droplet temperature. Herein, we describe the design of MINCZ, which fulfils the following requirements: (i) high accuracy and precision in measuring droplet temperatures within 0.2 K; (ii) ability to reach the homogeneous freezing point of pure water, with a median freezing temperature of 237.3±0.1 K for droplets with a diameter of 75 μm; and (iii) the ability to simultaneously perform several freeze–thaw cycles on hundreds of droplets. These characteristics allow to narrow the reported spread in nucleation rates as a function of temperature in past work, to detect mediocre and poor ice-nucleating particles at any temperature above that of homogeneous freezing, and to investigate the stochastic behaviour of nucleation. We validate MINCZ by measuring homogeneous freezing temperatures of water droplets and heterogeneous freezing temperatures of aqueous suspensions containing microcline, a common and effective INP in the atmosphere. In the future, MINCZ will be used to investigate the stochastic and deterministic behaviour of INPs, motivated by a need for better-constrained parameterizations of ice nucleation in weather and climate models, where the presence or absence of ice influences cloud optical properties and precipitation formation.



## 1 Introduction

Water in mixed-phase clouds is present in both the liquid and crystalline form, and the proportion between cloud droplets and ice crystals alters cloud radiative properties as well as cloud lifetimes (Lohmann, 2017; Lohmann and Feichter, 2005; Matus and L'Ecuyer, 2017). The transformation of liquid to ice in the troposphere can occur via homogeneous nucleation (in a pure water or aqueous droplet) or heterogeneous nucleation (for example, in a droplet containing solid particles). While homogeneous freezing of supercooled water occurs at temperatures below about 238 K, depending on droplet size and relative humidity (Ickes et al., 2015; Koop et al., 2000; Kreidenweis et al., 2018), heterogeneous nucleation in mixed-phase clouds may occur at temperatures up to 273 K in aqueous droplets containing impurities (ice-nucleating particles, INPs) that catalyse ice formation. Conversely, the presence of salt ions in solution may lead to a freezing point depression below the corresponding pure-water homogeneous or heterogeneous freezing temperature (Koop et al., 2000; Zobrist et al., 2008). A number of INP types are known to originate from natural and anthropogenic sources, including minerals such as feldspars, clay minerals, organic macromolecules, and organic matter (Kanji et al., 2017). However, the exact roles of the stochastic (time-dependent) and deterministic (time-independent) contributions to heterogeneous ice nucleation are uncertain and necessitate further research (Kaufmann et al., 2017; Knopf et al., 2020; Wright and Petters, 2013). A better understanding of these processes could improve our understanding of the role of INPs in precipitation formation so that present uncertainties in climate projections and weather forecasts may be reduced. In fact, the role of INPs in aerosol–cloud interactions has recently been identified as a research priority in the atmospheric community (Murray et al., 2021). Beyond the atmosphere, a more complete knowledge of ice nucleation is also pertinent to applications such as cryopreservation (Marquez-Curtis et al., 2021; Pegg, 2015) and pharmaceutical manufacturing (Assegehegn et al., 2019; Deck et al., 2022).

A range of techniques has been developed to study homogeneous and heterogeneous nucleation in atmospherically relevant systems (Diehl et al., 2014; Kaufmann et al., 2016; Miller et al., 2021; Rogers, 1988; Stetzer et al., 2008), and each technique can be associated with a particular drawback. For example, single-particle levitation devices (Diehl et al., 2014; Krämer et al., 1996) are time-consuming for investigating a large number of droplets sufficient for statistical analysis, whereas differential scanning calorimetry measurements of water-in-oil emulsions typically give only qualitative insight into nucleation behaviour due to the polydispersity in droplet size (Kaufmann et al., 2016; Klumpp et al., 2022; Kumar et al., 2018). To overcome such shortcomings, microfluidic techniques can be used to generate a stable, monodisperse population of water droplets at high throughput, suitable for quantifying nucleation rates. Water-in-oil emulsions are generated at an orifice, where the oil phase cleaves off the water phase to generate a droplet. Nonionic surfactants dispersed in the oil phase stabilize the droplets at the oil–water interface. At the microfluidic size scale, it becomes possible to investigate homogeneous ice nucleation, low INP concentrations, and INPs with mediocre or poor activity. Moreover, since microfluidic systems allow for the high-throughput generation of water-in-oil droplets, the number of droplets studied with this technique outnumbers the standard 96-well plates employed in many traditional droplet-freezing assays (e.g., David et al. (2019), Schneider et al. (2021), Garcia et al. (2012), and Kunert et al. (2018); see Miller et al. (2021) for a full list). Briefly, we note that the term cloud droplet denotes diameters up to approximately 50 μm in atmospheric science, while in microfluidics, a droplet can refer to larger sizes up to the nL range; hereafter, we refer to droplets more generally, not restricted to cloud droplet sizes.




Amongst existing microfluidic platforms designed for studying ice nucleation, there are two common approaches
for droplet generation and cooling: dynamic flow-through devices (Roy et al., 2021a; Stan et al., 2009; Tarn et al.,
2020, 2021) and static droplet arrays (Brubaker et al., 2019; Edd et al., 2009; Reicher et al., 2018; Roy et al.,
2021b). The flow-through approach is beneficial for analysing high numbers of droplets (between $10^3$ and $10^4$
(Tarn et al., 2020)) and therefore is particularly suitable for detecting low concentrations of INPs suspended in
water or an aqueous solution. Continuous flow devices are also desirable for potential use as autonomous in-line
instruments for monitoring the temporal evolution of INP concentration in the field (Tarn et al., 2020). One
drawback of current flow-through devices is the difficulty in independently controlling the cooling rate of droplets
over orders of magnitude. This is due to the fact that cooling rates are a function of fluid flow rate and channel
length, and changing these variables will also affect droplet diameter. A second drawback associated with
continuous flow devices is the inability to perform refreeze experiments on the produced droplets. On the other
hand, static droplet arrays are not suitable for detecting rare INPs in solution since such arrays generally only
contain between $10^2$ and $10^3$ droplets per experiment, and it is statistically unlikely for a rare INP to be present in
such a small volume of liquid (Brubaker et al., 2019; Reicher et al., 2018). Droplet arrays are beneficial in that
they can be cooled at various rates in a controllable fashion, providing the option of multiple cooling and thawing
cycles to gain insight into the stochastic vs. deterministic behaviour of heterogeneous ice nucleation.

In both flow-through and droplet array designs, microfluidic devices are almost always fabricated from
polydimethylsiloxane (PDMS) and plasma bonded to glass slides. PDMS is a hydrophobic, non-porous and gas-
permeable material. This gas permeability, however, can lead to the rapid evaporation and concomitant shrinking
of water droplets, limiting refreezing experiments.. Droplet evaporation can be reduced with various surface
treatments (Brubaker et al., 2019) or a blocking layer of a different material (Heyries et al., 2011), but to
permanently prevent gas permeation, alternative substrate materials must be considered. One alternative strategy
is to cool droplets off-chip on a solid substrate while covering them with a fluid of low gas-permeability like
silicone oil or squalene (Peckhaus et al., 2016; Wright and Petters, 2013). A second alternative is to store droplets
off-chip in tubing and immerse the tubing in an ethanol bath for cooling, as shown by Atig et al. (2018). It should
be noted that, in this study, droplet diameters were more than 1 mm, with the median freezing point of water at
this size being 249 K ($-24$ °C), i.e., far above homogeneous ice nucleation temperatures.

In cold-stage microfluidic platforms, droplets are typically cooled from below. Such an approach takes advantage
of the excellent heat transfer that accompanies miniaturisation, yet it is hampered by the poor heat transfer through
PDMS, which gives rise to a temperature gradient within the microfluidic device (Polen et al., 2018). Therefore,
measuring the actual temperature of droplets within the device remains a challenge, since cooling a microfluidic
device directly from the bottom generates a temperature gradient within the device. To account for such
temperature differentials, Reicher et al. (2018) calibrated droplet temperatures as a function of cold-stage
temperature by observing the melting of solutions and materials with known melting points. As discussed by
Reicher et al. (2018), a different calibration equation was needed for each PDMS substrate thickness, which was
identified by Polen et al. (2018) as a potential drawback. To avoid a thickness-dependent calibration, Tarn et al.
(2020, 2021) placed a thermocouple within a microfluidic channel parallel to the one through which droplets flow



to more accurately determine droplet temperature, but the reported uncertainty in this setup is still at a relatively
high value of $\pm 0.7$ K. Given that uncertainties in homogeneous ice nucleation rates are dominated by uncertainties
in temperature (Riechers et al., 2013), increasing an instrument's temperature accuracy is the single most
important factor in improving our ability to precisely discern how nucleation rate changes as a function of
temperature. This is especially important because nucleation rates for the homogeneous freezing of water obtained
from various instrument types (continuous flow chambers, droplet freezing assays, etc.) and instruments of the
same type (e.g., all microfluidic platforms) currently span several orders of magnitude at the same temperature
(Ickes et al., 2015; Tarn et al., 2021).
Amongst the rapidly-growing number of microfluidic systems designed to investigate ice nucleation, we aimed
to develop a setup able to create and freeze picoliter-sized droplets, whilst avoiding the primary disadvantages
associated with current methods. Namely, our goals were to achieve a monodisperse size distribution of droplets
with diameters of 75 μm, generate a large number of droplets (many hundreds), ensure droplet stability over the
time needed to perform multiple (re-)freezing cycles at various cooling rates, minimize temperature gradients in
the device, and ensure high accuracy and precision in all temperature measurements. Further, and most
importantly, we aimed to develop a system that is easy to handle and easy to transfer to other laboratories or field
sites. Herein, we present and validate our system and technique. We report data for the homogeneous freezing of
pure water and for the heterogeneous freezing of microcline suspensions in water. Microcline, a K-feldspar, is
selected as an example, since it is commonly found in collected mineral dust samples and it is a highly active INP
(Harrison et al., 2016; Kanji et al., 2017; Klumpp et al., 2022; Welti et al., 2019).

## 2   Materials and Methods

In the Microfluidic Ice Nuclei Counter Zürich (MINCZ), droplets are generated in a conventional PDMS
microfluidic device. Droplets are not stored on-chip, but in fluorinated (perfluoroalkoxy alkane, PFA) tubing
having an inner diameter of 75 μm. The PFA tubing is immersed and cooled in an ethanol bath, minimizing any
temperature gradients, while maximizing heat transfer. The chemically inert and relatively gas-impermeable PFA
tubing allows for prolonged cooling cycles and refreeze experiments to temperatures below which pure water
freezes homogeneously. A CMOS camera connected to a stereoscope is used to image the droplets and a semi-
automated image analysis algorithm is used to identify droplet freezing events. We present a general summary of
the components that comprise MINCZ (Sect. 2.1), followed by detailed descriptions of the microfluidic chip
(Sect. 2.2) and aqueous sample preparation (Sect. 2.3). Finally, the workflow of a typical experiment is presented,
including droplet generation (Sect. 2.4.1), droplet cooling (Sect. 2.4.2), and image analysis to determine droplet
size (Sect. 2.4.3) and freezing temperature (Sect. 2.4.4).

### 2.1  Instrument design

Figure 1 presents an overview of the equipment used in MINCZ. Each piece of equipment is categorized based
on its function, whether it is used during droplet generation (Fig. 1a and 1c) or droplet cooling (Fig. 1b and 1d).
A stereoscope (Nikon SMZ1270, 0.5× objective lens, fibre ring illuminator with LED light source) connected to
a CMOS camera (iDS UI-3060CP-M-GL Rev. 2) is used in both steps to obtain images. For droplet generation





(see Sect. 2.4.1 for more details), we use: i) three syringe pumps fitted with 1 mL glass syringes; ii) a PDMS
microfluidic chip; and iii) high-purity perfluoroalkoxy alkane (PFA) tubing that is directly inserted into the outlet
of the microfluidic chip and kept in place in a custom-milled polyether ether ketone (PEEK) holder. For droplet
cooling (see Sect. 2.4.2 for more details), we use: i) an ethanol cooling bath (insulated by a custom 3D-printed
structure) to immerse the droplet-containing PFA tubing; ii) two K-type thermocouples; iii) a Peltier element
connected to a power supply and cooled from below by a heat transfer fluid circulating through an aluminium
block connected to a chiller. To improve image quality during droplet cooling, we use a pair of cross-polarized
filters, and we place six glass cover slips underneath the PEEK tubing holder for improved image contrast.

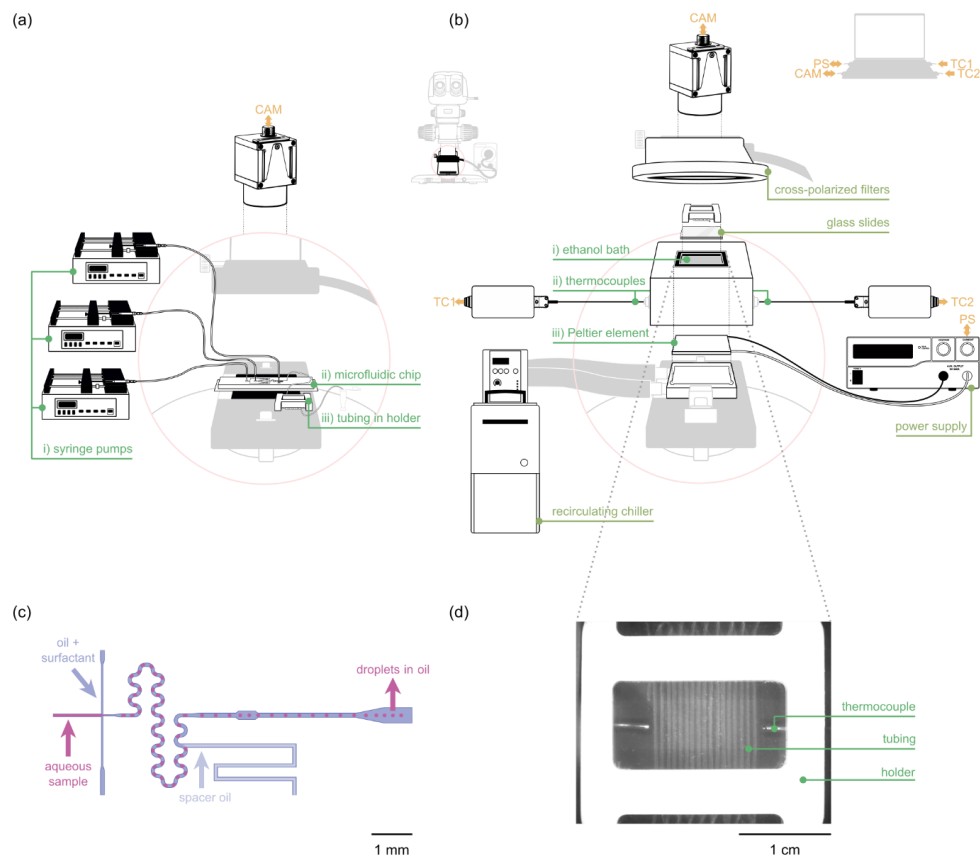

**Figure 1. Overview of the Microfluidic Ice Nuclei Counter Zürich (MINCZ) equipment grouped into (a) the droplet**
**generation step with (i) syringe pumps, (ii) a microfluidic chip, and (iii) PFA tubing in a PEEK holder; and (b) the**
**droplet cooling zone with (i) an ethanol bath, (ii) two thermocouples, and (iii) a Peltier element. (c) A schematic of the**
**microfluidic channels used to generate aqueous droplets surrounded by an oil–surfactant continuous phase. (d) A**
**top-down image of the ethanol bath into which the PEEK holder with PFA tubing is placed.**
**2.2  Microfluidic chip design and fabrication**
The microfluidic chip design was drawn in AutoCAD® 2018 (Autodesk, San Rafael, USA). It features a flow-
focusing droplet generator with an orifice that is 75 µm high and 20 µm wide. After passing through passive-



mixing structures, the droplets flow from a 350 µm wide outlet into the 75 µm inner diameter PFA outlet tubing.
A schematic representation is shown in Figure 1c. The chip design was printed onto a high-resolution film
photomask (Micro Lithography Services Ltd, Chelmsford, UK) which was used to pattern an SU-8 (GM1070,
Gersteltec, Switzerland) coated silicon wafer (10 mm diameter, 525±25 thickness, <100> orientation, Siegert
Wafer GmbH, Germany). This resulting master mould was employed to fabricate the PDMS chips by pouring
PDMS (Elastosil RT 601 A/B, Ameba AG, Switzerland) over the mould at a 10:1 mass ratio of base to curing
agent, with subsequent curing at 70 °C for more than two hours. Inlets (0.76 mm) and outlets (0.41 mm) were
punched with a hole-puncher (Shafts 20 and 25, Syneo, USA), and the PDMS devices were plasma bonded
(plasma cleaner, Diener electronic GmbH, Germany) to planar glass slides (Menzler-Glaser, Germany). To
improve hydrophobicity, the PDMS devices were incubated in 5 % v/v (tridecafluoro-1,1,2,2-
tetrahydrooctyl)trichlorosilane (97 %, abcr GmbH, Germany) for 5 minutes, then in HFE-7500 (3M™ Germany)
for 5 minutes, and then kept on a hot plate at 120 °C for at least 14 hours.
**2.3  Sample preparation**
For the homogeneous freezing assays, ultrapure water was used (molecular biology reagent-grade, 0.1 µm filtered,
Sigma–Aldrich, USA), hereafter referred to as Sigma–Aldrich (SA) water. The microcline used in the
heterogeneous ice nucleation experiments was from the same milled stone from Elba, Italy, as reported in a
previous study (Welti et al. (2019); for mineralogical composition, see X-ray diffraction results therein). Scanning
electron microscopy (SEM) revealed a high size-polydispersity of the mineral particles ranging from sub-
micrometer to more than 30 µm (Fig. A1a). Indeed, individual particles were clearly visible when suspended in
microfluidic droplets (Fig. A2). To ensure repeatability and reproducibility, we homogenized the microcline to
particles in the sub-micrometer range using the following procedure. First, the mineral sample (2 g in 50 mL SA
water) was sonicated (8 × 30 s pulse in a UP200ST ultrasonic VialTweeter (Hielscher Ultrasonics GmbH,
Germany)) followed by filtration using a 0.45 µm polyethersulfone sterile syringe filter (TPP Techno Plastic
Products AG, Switzerland). Then, the resulting homogeneous mineral sample was concentrated and dried using a
SpeedVac (Savant™ SPD111V, Thermo Scientific™, USA). Just before use, the resulting pellet of mineral
particles was rehydrated to a stock solution of 1.5 mg mL$^{-1}$ in SA water, and this stock solution was subsequently
diluted to the working solution of 0.5 mg mL$^{-1}$ and sonicated in a water bath for 15 minutes. The size distribution
of the microcline particles was visualized using scanning electron microscopy (SEM; FEI Magellan 400 Scanning
Electron Microscope), as shown in Fig. A1c.
**2.4  Experimental workflow**
Figure 2 summarizes the workflow of an experiment using MINCZ. Spherical water-in-oil droplets are generated
within a PDMS chip (see Sect. 2.4.1 for details) and introduced into the PFA tubing. A video is recorded during
droplet generation, from which the mean droplet diameter can be evaluated (see Sect. 2.4.3). Afterwards, the
droplet population within the PFA tubing is cooled in the ethanol bath, while images are captured at a frequency
sufficient to obtain one image for every 0.05 K decrease in temperature, depending on the user-specified cooling
rate (see Sect. 2.4.2). We process the saved images using a semi-automated image analysis algorithm to determine
the number of frozen droplets as a function of temperature (see Sect. 2.4.4).



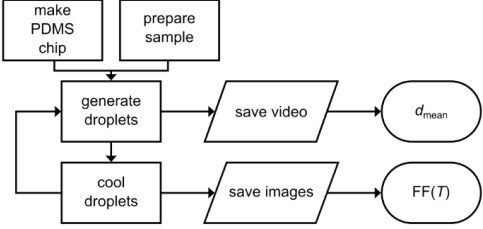

**Figure 2.** Workflow of an experiment using MINCZ consisting of PDMS chip fabrication and sample preparation,
**followed by droplet generation and cooling, where a high-speed video is taken to determine mean droplet diameter**
**and a series of images are taken to determine the frozen fraction (FF) of droplets as a function of temperature.**

### 2.4.1 Droplet generation

As seen in Figure 1a and 1c, the PDMS microfluidic chip is connected to two pieces of PTFE tubing (0.56 mm
ID, 0.25 mm OD, Rotima AG Switzerland) containing the water phase and the surfactant in oil (5 % 008-
FluoroSurfactant (RAN Biotechnologies, USA) diluted to 1 % v/v in HFE-7500) for droplet generation, while a
third piece of tubing containing fluorinated oil (HFE-7500) is employed as a spacer fluid. Glass syringes (1 mL
Hamilton® syringe, Sigma–Aldrich, USA) are filled with a supporting fluid (either water or fluorinated oil) and
held in syringe pumps (Aladdin AL1000-220Z, World Precision Instruments, USA), which are employed to ensure
stable flow rates. An air bubble between each injected fluid (the aqueous sample and the surfactant–oil mixture)
and the supporting fluid in the PTFE tubing prevents contamination and dilution of the sample by the supporting
fluid, whilst allowing for flexible and low sample consumption. One end of the PFA tubing for droplet storage
(50 cm in length, 360 μm OD, 75 μm ID, IDEX Health & Science LLC, USA) is directly inserted into the PDMS
device outlet. The rest of the tubing is kept in the custom-milled PEEK holder. During droplet generation, the
PDMS device is monitored using the stereoscope and camera. After a stable generation of spherical droplets is
achieved and a video of droplet generation is recorded, the PFA tubing is immediately cut from the PDMS chip
with scissors, and the tubing ends mechanically blocked using tweezers.

The flowrates used in the current study are listed in Table 1 for the SA water experiments and Table 2 for the
microcline experiments. The same PDMS chip can be reused for several consecutive runs (e.g., for the generation
of the three microcline suspensions in Table 2), or separate chips may be used if channels become clogged between
experiments or if the chip delaminates from the glass slide due to insufficient bonding (e.g., in Table 1). As a
result of new chips being used from one day to another, the flow rates in Table 1 and Table 2 required for stable
droplet generation differ slightly.





**Table 1: Sigma–Aldrich (SA) water, surfactant in oil, and spacer oil flowrates used to produce each population of**
**droplets for the homogeneous freezing experiments. The mean diameter of droplets obtained from the captured high-**
**speed video is also summarized for each droplet population.**

|  | $Q_{water}$ [μL min$^{-1}$] | $Q_{surfactant}$ [μL min$^{-1}$] | $Q_{spacer oil}$ [μL min$^{-1}$] | $d_{mean}$ [μm] |
|---|---|---|---|---|
| day 1 | 1.0 | 1.5 | 2.0 | 75 ± 5 |
| day 2 | 1.0 | 1.5 | 2.3 | 75 ± 5 |
| day 3 | 1.0 | 2.0 | 1.4 | 78 ± 5 |


**Table 2: Microcline suspension, surfactant in oil, and spacer oil flowrates used to produce each population of droplets**
**for the heterogeneous freezing experiments. The mean diameter of droplets obtained from the captured high-speed**
**video is also summarized for each droplet population.**

|  | $Q_{microcline}$ [μL min$^{-1}$] | $Q_{surfactant}$ [μL min$^{-1}$] | $Q_{spacer oil}$ [μL min$^{-1}$] | $d_{mean}$ [μm] |
|---|---|---|---|---|
| i | 0.8 | 1.5 | 2.3 | 78 ± 5 |
| ii | 0.8 | 1.5 | 2.3 | 73 ± 5 |
| iii | 0.9 | 1.5 | 2.3 | 73 ± 5 |

**2.4.2    Droplet cooling**
The PFA tubing containing the droplets is immersed in an ethanol bath held in an aluminium container (40 mm ×
40 mm × 60 mm). The inside walls of the bath are oxidized to provide a black background behind the droplets to
improve imaging contrast. Six glass cover slips (24 mm × 24 mm, 0.13–0.17 mm thick, Fisherbrand™, Fisher
Scientific AG, Switzerland) are placed under the PFA tubing to further improve contrast. To ensure that
temperature measurements are representative of actual droplet temperatures, two thermocouples (K-type, 0.5 mm
OD, RS Components GmbH, Germany, and TC Direct, Germany) are placed horizontally in the ethanol bath in
the same plane as the PFA tubing (Fig. 1b), with the average of the recorded temperatures taken to be
representative of the temperature of the droplets. Each thermocouple was calibrated to the melting point of
mercury (−38.8 °C or 234.4 K) and water (0 °C or 273.15 K), providing a high accuracy with a standard deviation
of 0.1 K for three measurements at each melting point. Over all experiments reported herein, the average difference
in the measured temperature between the two thermocouples ($T_2 - T_1$) in the ethanol bath was 0.01 ± 0.21 K
(standard deviation). The uncertainty in our temperature measurement is thus reported to be ± 0.2 K.

A Peltier element (PKE 128A 0020 HR 150, Peltron GmbH, Germany) is connected to a laptop-controlled power
supply (Manson® HCS-3302, Distrelec Group AG, Switzerland) to achieve the user-defined cooling rate. Heat
from the Peltier element is dissipated from below by an aqueous 55 % v/v ethylene glycol (98 % technical grade,
Sigma–Aldrich, USA) mixture circulating through an aluminium block connected to a chiller (Huber KISS K6,
Huber Kältemaschinenbau AG, Germany). Thermal paste (Fischer Elektronik GmbH, Germany) is applied
between the top of the aluminium block and the bottom of the Peltier element to ensure good thermal contact.




A custom Python-based (Python 3.0) user interface was designed to permit the user to select the desired cooling
rate and image acquisition settings. Once these parameters are selected and the temperature of the ethanol bath
has reached steady state (with the chiller set to −15 °C and the power supply at 0.8 V), cooling is initiated. A
proportional controller with a temperature-dependent gain parameter sets the voltage of the power supply to
maintain this cooling rate (see Figs. B1 and B2 for the time series of cooling rate as a function of temperature for
each experiment reported herein). During cooling at 1 K min$^{-1}$, images are captured every three seconds, and the
temperature is recorded. Once the measured temperature reaches the set end temperature, e.g. 233 K, the program
terminates.

### 2.4.3    Droplet size distribution

From a 10 second video of droplet generation, the mean droplet size is determined through a series of image
processing steps implemented in Python (using the cv2 and skimage packages): subtracting the background,
equalizing the histogram, morphological opening, thresholding, and using the Hough circle transform to identify
and measure the droplets in each frame of the captured video. The obtained mean diameter for each droplet
population is summarized in Table 1 and Table 2 for pure water and microcline suspensions, respectively. The
uncertainty in mean diameter is estimated to be ± 5 μm (corresponding to an uncertainty of 2 pixels in the droplet
radius).

### 2.4.4    Freezing detection

Due to the high purity of the SA water, only a weak increase in brightness is detected when a droplet freezes (i.e.,
the raw change in pixel intensity between the background and an unfrozen droplet vs. a frozen droplet is minimal),
possibly because few impurities are present to induce crystallographic defects that manifest as an increase in
brightness. Therefore, when combined with a low number of pixels per droplet, the detection of droplet freezing
in the saved images is challenging and necessitates a semi-automated approach.

An overview of the workflow for detecting droplet freezing is illustrated in Figure 3. If necessary, prior to
automated screening, an image stabilization routine is applied to the images using the cv2 and skimage packages
in Python for feature detection and Euclidian transformation. To detect droplet freezing, the images are first
automatically screened to find locations where droplet freezing may have occurred. Second, the user is prompted
to classify whether freezing did or did not occur. In the future, the manually-labelled images of frozen or unfrozen
droplets could be used to train a machine learning algorithm for fully-automated image processing. Droplets that
exhibit a clear spike in brightness upon freezing would facilitate the automation of image classification. A distinct
brightness change is expected for droplets containing solid impurities, such as INPs, or aqueous solutions of, for
example, NaCl.

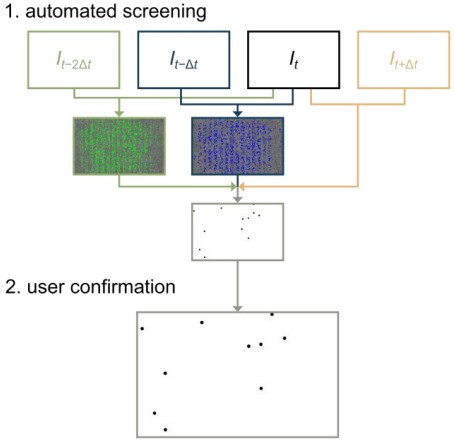

**Figure 3. Workflow to locate the droplets that froze between two consecutive images ($I_t$ and $I_{t-\Delta t}$), also making use of images $I_{t-2\Delta t}$ and $I_{t+\Delta t}$. The first step is to automatically screen potential locations where a droplet may have frozen, by comparing the brightness change in a location between two consecutive images $I_t$ and $I_{t-1\Delta t}$ (potential freezing events highlighted with blue pixels). During this first step, we also screen for false positives due to droplet motion or impurities in the ethanol bath by analyzing additional images $I_{t-2\Delta t}$ (potential freezing events highlighted with green pixels) and $I_{t+\Delta t}$ for brightness changes at the same pixel coordinate. The second step is for the user to confirm whether a droplet actually froze at that location (to eliminate false positives due to noise or other optical interference).**

The automated screening procedure includes multiple steps: subtracting the pixel intensities of two consecutive images taken at time $t$ and $t - \Delta t$, applying a bilateral filter to the subtracted image, carrying out morphological opening, detecting edges, and applying a Hough circle transform to find potential droplet centres. To reduce the number of potential droplets that must be classified by the user, the above procedure is always performed for two pairs of images, resulting in the difference images $I_{-\Delta t} = I_t - I_{t-\Delta t}$ (with potential droplet centres highlighted in blue in Figure 3) and $I_{-2\Delta t} = I_t - I_{t-2\Delta t}$ (with potential droplet centres highlighted in green in Figure 3). Only those coordinates where brightness changes are detected in both image pairs are considered as potential freezing events. Additionally, two criteria were defined that must be met in the $I_{-\Delta t}$ image to definitively tag a droplet: (i) the identified coordinate must fall within a predefined grid of pixels where tubing is present; (ii) the average pixel intensity of an 8-pixel radius at that coordinate must be less than 90 (i.e., dark in the range of grayscale values between 0 and 255). Finally, the average pixel intensity of an 8-pixel radius at that coordinate in the $I_{+\Delta t} = I_{t+\Delta t} - I_t$ image must be less than 150. The user can also flag any frozen droplets that are not spherical as a result of two droplets coalescing. These frozen droplets with twice the volume are discarded from further analysis.





## 3 Results and Discussion

Figure 4 depicts the fraction of frozen droplets as a function of temperature for three independent droplet populations of Sigma–Aldrich (SA) water cooled at a rate of 1 K min$^{-1}$. After being frozen once, the third droplet population was thawed and refrozen twice more (day 3b and 3c). In each frozen fraction curve, there is a single data point corresponding to each saved image (that is, one data point at every interval of 0.05 K showing the cumulative number of droplets frozen down to that temperature). From the three independent droplet populations, the median freezing temperature is reproducible within a narrow temperature range of 237.3 ± 0.1 K (standard deviation). Possible contributions to the observed variability could arise from inherent uncertainty in the thermocouple measurement, small changes in the positioning of the tubing holder and thermocouples between experiments, and/or slight differences in droplet diameter between droplet populations. The repeated freeze–thaw cycles yield an even narrower median temperature range of 237.41 ± 0.04 K (standard deviation), a variability that can be attributed solely to inherent uncertainty in the thermocouple measurement, because there were no changes to the droplet population or to the positioning of the holder or thermocouples. Given the high reproducibility of results over three freezing cycles, MINCZ is ideally suited for investigating questions surrounding the stochasticity of nucleation in a single droplet, in contrast to continuous flow microfluidic devices that are well-suited for high-throughput analysis when detecting the presence of rare ice-nucleating particles. For comparison, Figure 4 also shows the frozen fraction calculated based on the recommended parameterization for the homogeneous nucleation rate of water from Ickes et al. (2015) (see Appendix C for more details), which was obtained by fitting to a wide range of previously-reported experimental data and is representative of current state-of-the-art. The frozen fractions observed using MINCZ are in general agreement with this parameterization. The accurate and reproducible results for the median freezing temperature of pure water droplets and the lack of an early freezing onset validates MINCZ as a reliable instrument that can be used to detect freezing due to mediocre ice-nucleating particles at any temperature above the onset of homogeneous ice nucleation. Early freezing onset can occur due to impurities present in the pure water sample that would appear, for example, as a slow increase in frozen fraction at higher temperatures, as seen in the freezing behaviour of pure water in Peckhaus et al. (2016) and Brubaker et al. (2019). The ability of MINCZ to reach such low temperatures is achieved with very low droplet volumes (approx. 200 pL) and the absence of a solid substrate that may initiate the nucleation of ice at higher temperatures. Lastly, we confirmed that there is no spatial bias in freezing behaviour across the observed area, as summarized in Appendix B.



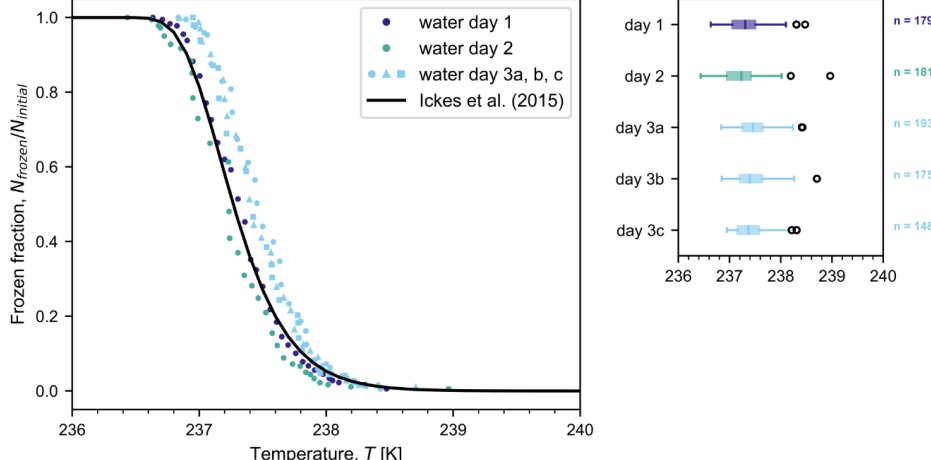

**Figure 4. Frozen fraction of pure water (Sigma–Aldrich) droplets (with diameters as indicated in Table 1) as a**
**function of temperature for different droplet populations (with *n* number of droplets) cooled at a rate of 1 K min$^{-1}$ on**
**three separate days. The droplet population on day 3 was subjected to three freeze–thaw cycles (a, b, c). Also shown is**
**the frozen fraction curve for the homogeneous freezing of water using the parameterization from Ickes et al. (2015)**
**for droplets with a diameter of 75 µm. Boxplots on the right-hand side summarize the experimental results. The**
**center line of each boxplot indicates the median freezing temperature, the box spans the interquartile range (between**
**the 25$^{th}$ and 75$^{th}$ percentiles), the whiskers extend to the maximum and minimum temperatures, and outliers are**
**shown by open circles. The temperature uncertainty of our measurements is estimated to be ± 0.2 K.**
Figure 5 shows the frozen fraction of droplets as a function of temperature for aqueous suspensions containing
0.05 wt % microcline, also cooled at a rate of 1 K min$^{-1}$. Three independent droplet populations were generated
and cooled, yielding a median freezing temperature of 244.6 ± 0.7 K. As in Figure 4, one data point is plotted for
every 0.05 K interval in temperature, showing the cumulative number of droplets frozen down to that temperature.
In comparison to the results for pure water, droplets containing microcline particles froze at higher temperatures
and over a wider range. Additionally, the standard deviation of the median freezing temperature increased,
showing a higher variability between runs. This widening of freezing temperature and increase in variability
relative to that seen for homogeneous freezing can be explained by inherent variations in the amount and activity
of the mineral particles present in each droplet. As investigated by Knopf et al. (2020), variations in the surface
area of the mineral in each droplet can be one source of variability in the frozen fraction. In Figure 5, we also
show results reported by Welti et al. (2019) using the same microcline sample, but studied using the Zurich Ice
Nucleation Chamber (ZINC) with particles size-selected to a mobility diameter of 400 nm or 800 nm. Finally, in
Figure 5, we also include the frozen fraction of water droplets (~750 droplets with volumes of 0.2 nL) containing
0.05 wt % microcline (sample named FS02) printed onto a solid substrate and cooled at 1 K min$^{-1}$ by Peckhaus
et al. (2016). Both mineral samples were predominantly microcline (~90 % K-feldspar and ~10 % Na-feldspar in
Welti et al. (2019); 80 % K-feldspar, 16 % Na/Ca-feldspar, and 4 % quartz in Peckhaus et al. (2016)). Overall, the
frozen fraction curves obtained from MINCZ and ZINC show ice nucleation activity of the microcline particles
in a similar temperature regime, with freezing in MINCZ occurring at temperatures close to those of the 400 nm
particles in ZINC; all of these frozen fraction curves are at lower temperatures compared to the data obtained by
Peckhaus et al. (2016).

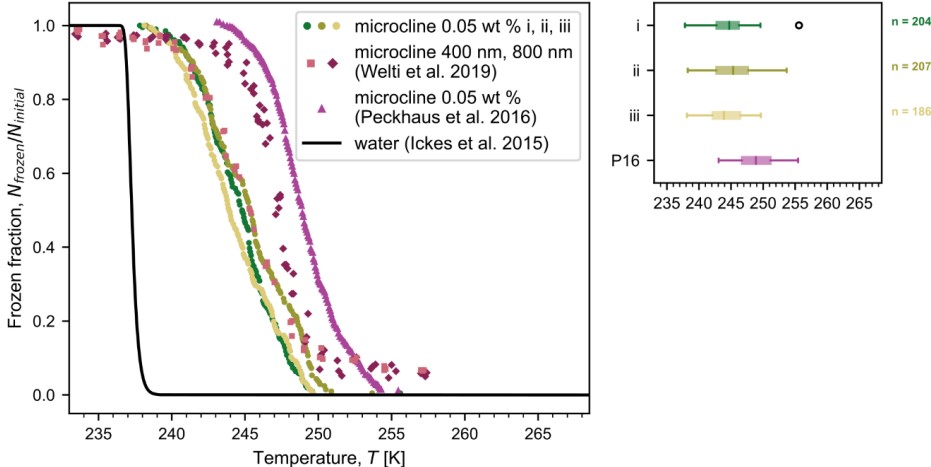


**Figure 5. Frozen fraction of microcline (0.05 wt % in SA water) droplets (with diameters as indicated in Table 2) as a function of temperature for three independent droplet populations (i, ii, iii with *n* number of droplets) cooled at a rate of 1 K min⁻¹. For comparison, we show experimental results reported in Welti et al. (2019) obtained with the same microcline sample but using the Zurich Ice Nucleation Chamber (ZINC) for particles size-selected to a mobility diameter of 400 nm or 800 nm. The frozen fraction curve digitized from Peckhaus et al. (2016) (P16 in the boxplot) is also shown for comparison, where 0.2 nL aqueous droplets with 0.05 wt % microcline suspension were printed onto a solid substrate and cooled at 1 K min⁻¹. Also illustrated is the frozen fraction curve for the homogeneous freezing of water using the parameterization from Ickes et al. (2015) for droplets with a diameter of 75 μm. On the right, a boxplot compares the freezing temperatures of the three droplet populations, where the center line indicates the median freezing temperature, the box spans the interquartile range (between the 25th and 75th percentiles), the whiskers extend to the maximum and minimum temperatures, and outliers are shown by open circles. The temperature uncertainty of our measurements is estimated to be ± 0.2 K.**

We note that the curves obtained using MINCZ depend on the concentration of microcline in suspension, since
any change to the available surface area will shift the observed temperature of ice nucleation accordingly. For our
concentration of 0.05 wt%, the expected surface area is on the order of $10^{-10}$ m² (assuming a Brunauer–Emmett–
Teller (BET) adsorption specific surface area between 1.9 m² g⁻¹ (Atkinson et al., 2013) and 3.2 m² g⁻¹ (Kumar et
al., 2018)). In contrast, single particles were investigated using ZINC with surface areas on the order of $10^{-13}$ to
$10^{-12}$ m² for 400 nm and 800 nm, respectively. Typically, median freezing temperatures increase as particle surface
areas increase (e.g., as seen in Welti et al. (2019)), because there is an increased probability that the surface
contains a nucleation site that is active at higher temperatures. Since the surface area of microcline per droplet in
MINCZ is at least two orders of magnitude larger than that of a single particle, it may be expected that the median
freezing temperature of these droplets would be at a higher temperature than the median freezing temperature of
droplets with a single particle in ZINC. However, we observe that the frozen fraction curves obtained with MINCZ
are in agreement with the 400 nm particles analysed in ZINC, but freeze at lower temperatures compared to the
800 nm particles analysed in ZINC. This could be explained by a mineralogical bias due to 450 nm filtration of
the solution used in MINCZ that shifts freezing towards lower temperatures. That is, the larger particles may
exhibit a higher density of active sites that induce freezing at higher temperatures because of a size-dependent
mineralogical composition or morphology, and as a result, increasing the surface area by increasing only the
number of sub-450 nm particles in the droplets would not increase the probability of nucleation. Alternatively, if



there was in fact no mineralogical bias depending on particle size, the activity of the microcline could have instead
decreased over its storage time as a dry sample over a period of seven years from when it was previously analysed
in ZINC.

Finally, we can compare the frozen fraction of microcline suspensions studied using MINCZ to that obtained by
Peckhaus et al. (2016), where the same microcline concentration was investigated (0.05 wt%) at the same cooling
rate of 1 K min$^{-1}$. The main difference between these two studies was in sample preparation: we sonicated and
filtered the microcline suspension prior to cooling, but the sample was only suspended in solution after milling
the stone sample in Peckhaus et al. (2016). Similar to the discrepancy in the frozen fractions between MINCZ and
ZINC, it is again not possible to determine why the observed frozen fraction is at lower temperatures compared
to the data in Peckhaus et al. (2016). Either there could have been a mineralogical bias due to 450 nm filtration,
or the activity of the microcline sample studied herein could have been lower than the activity of the sample
studied by Peckhaus et al. (2016). An inherent difference in ice nucleation activity of two microcline samples
collected at different locations has also been observed by Kaufmann et al. (2016), who investigated the same
sample from Elba as Welti et al. (2019) and a sample from Namibia. They found that the sample from Namibia
exhibited a higher ice nucleation activity than the one from Elba despite its lower microcline content.
**4    Conclusions**
The MINCZ platform employs microfluidic technology to generate homogeneously-sized droplet populations of
approximately 75 μm in diameter that are then cooled off-chip in PFA tubing immersed in ethanol. We presented
the validation of this technique for the homogeneous freezing of pure water as well as heterogeneous freezing
using microcline. Our obtained results in the temperature range of homogeneous freezing fit well within the
expected temperature ranges reported previously. By immersing the tubing containing the droplets in a cooling
bath, MINCZ cools the droplets from all directions, instead of only from below, reducing the temperature gradient
and therefore yielding a high temperature accuracy of 0.2 K. The lack of early-onset freezing events in our data
obtained for homogeneous nucleation indicates that there are very few, if any, impurities in the water used in this
work. Therefore, in future studies this characteristic allows the delineation between freezing due to the
homogeneous pathway and freezing due to mediocre or poor INPs that are only active at relatively low
temperatures. We showed that by storing droplets in gas-impermeable PFA tubing, multiple highly-reproducible
refreezing cycles can be performed. The semi-automated approach for freezing droplet detection allows for the
study of statistically high numbers of droplets (in excess of $10^2$) in parallel. Furthermore, the instrument is
comprised of simple components (e.g., stereoscope, Peltier element, chiller, and CMOS camera), and it has a
relatively small footprint in the lab. These attributes make MINCZ also suitable for transfer to other laboratories
or field sites. Future work will focus on further automation of the operation of MINCZ to ensure continued
reproducibility by limiting user-dependent influences.

**Appendix A: Microcline particle imaging**

Figure A1 shows secondary electron (SE) scanning electron microscopy (SEM) images of microcline suspensions
that were (a) untreated, (b) sonicated with $8 \times 30$ s pulses in an ultrasonic VialTweeter, and (c) sonicated followed
by filtration (0.45 µm polyethersulfone sterile syringe filter). Figure A2 shows images of microfluidic droplets
with untreated microcline suspensions at two concentrations (0.1 wt % and 2 wt %), where the heterogeneity in
microcline particle size is clearly visible. While sonication successfully broke apart the microcline particles, a
significant portion of larger particles remained (Fig. A1b). After sonication and filtration, the remaining particles
were more uniform in size (Fig. A1c).

(a) untreated

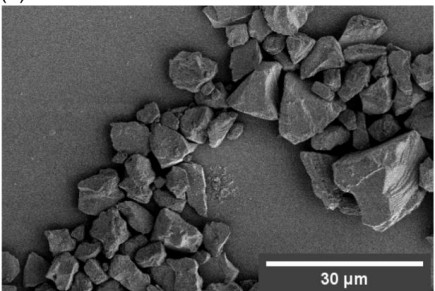

(b) sonicated

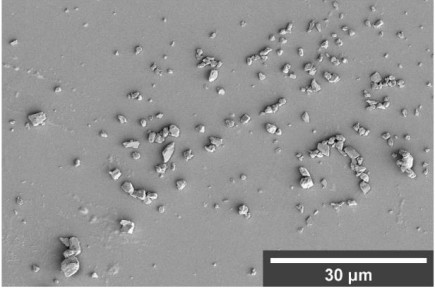

(c) sonicated and filtered

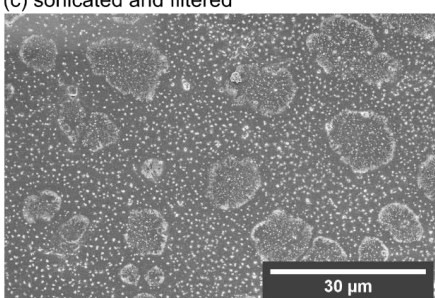

**Figure A1. Scanning electron microscopy images of microcline that was (a) untreated, (b) sonicated with $8 \times 30$ s pulses in an ultrasonic VialTweeter, and (c) sonicated using the same procedure as (b) but additionally filtered (0.45 µm syringe filter).**

(a)

(b)

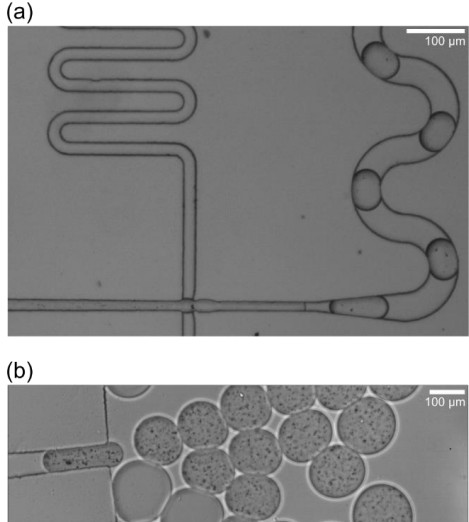

**Figure A2. Microfluidic droplets of aqueous suspensions containing (a) 0.1 wt % and (b) 2 wt % microcline that were**
**neither sonicated nor filtered. Microcline particles in these droplets are clearly visible as black pixels in both images.**
**The slight difference in droplet sizes can be accounted to partial clogging of the droplet generating orifice due to the**
**high concentration of large mineral particles in this particular experimental run.**



**Appendix B: Spatial distribution of freezing events and cooling rate for each experiment**

Figures B1 and B2 summarize the spatial temperature distribution of freezing events in the first two columns, where each symbol represents one droplet freezing at a specific temperature and *x*- or *y*-coordinate. Over all experiments (Fig. B1 for pure water and Fig. B2 for microcline suspensions), it is evident that there is no spatial bias in freezing behaviour. The third column of each figure shows the measured cooling rate over the course of each experiment, calculated based on the previous 60 s at each temperature where an image was saved (i.e., $\mathrm{d}T/\mathrm{d}t = (T(t) - T(t - 60\,\mathrm{s}))/(60\,\mathrm{s})$).

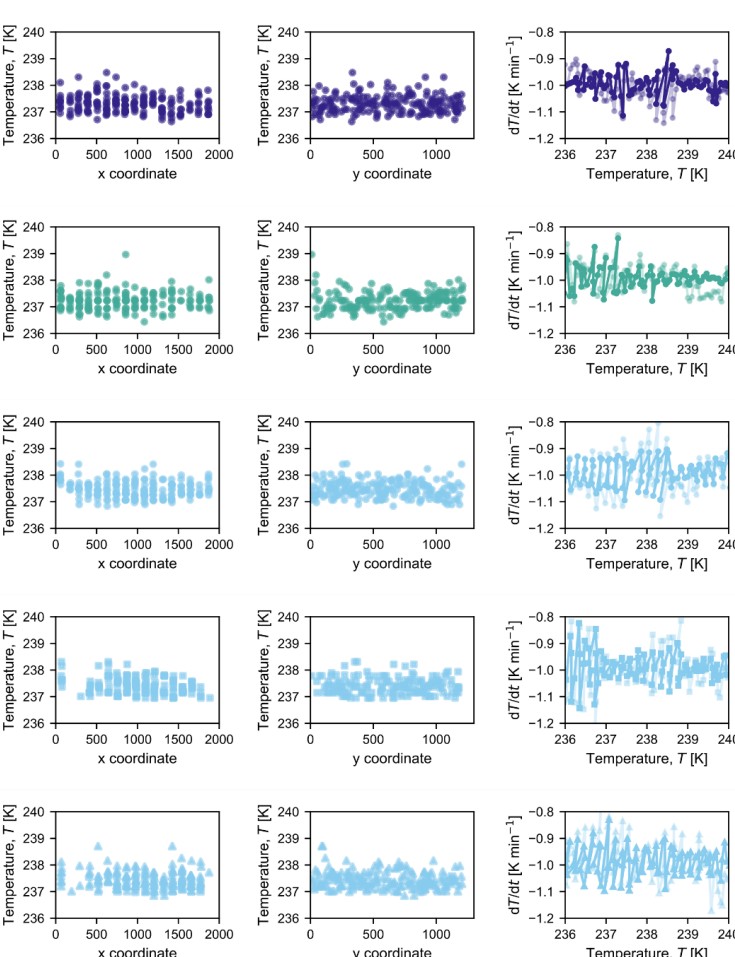

**Figure B1. Compilation of observed freezing temperatures at each *x*- and *y*- pixel location to illustrate that there is no discernable spatial bias in freezing temperature for each experiment conducted with pure water in Figure 4 (from top to bottom: water day 1, water day 2, and water day 3a, b, and c). The third graph in each row shows the measured cooling rate at each temperature where a picture was taken; the opaque line indicates the cooling rate measured by the thermocouple that was used as input to the control loop, and the semi-opaque line indicates the cooling rate measured by the second thermocouple in the bath.**

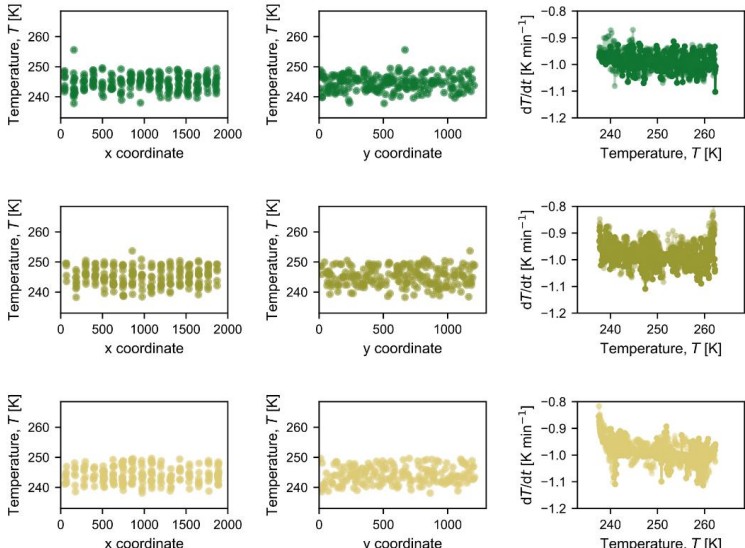

**Figure B2. Compilation of observed freezing temperatures at each *x*- and *y*- pixel location to illustrate that there is no discernable spatial bias in freezing temperature for each experiment conducted with the microcline suspension shown in Figure 5 (from top to bottom: i, ii, and iii). The third graph in each row shows the measured cooling rate at each temperature where a picture was taken; the opaque line indicates the cooling rate measured by the thermocouple that was used as input to the control loop, and the semi-opaque line indicates the cooling rate measured by the second thermocouple in the bath.**


**Appendix C: Calculation of frozen fraction from nucleation rate**

Following the derivation in Pruppacher and Klett (2010, p.211), the differential number of droplets that remains
unfrozen in a differential time can be integrated to yield

$$f_{\text{un}} = \frac{N_{\text{un}}}{N_0} = \exp(-V_{\text{d}} J_{\text{hom}} t) \tag{C1}$$

where $f_{\text{un}}$ is the fraction of droplets that remains unfrozen (where $N_{\text{un}}$ is the number of unfrozen droplets after
time $t$, and $N_0$ is the total number of unfrozen droplets at time $t = 0$), $V_{\text{d}}$ is the volume of a droplet, and $J_{\text{hom}}$ is
the homogeneous nucleation rate.

To evaluate our experiments, we count the frozen droplets at fixed time intervals, $\Delta t$. As we cool the droplets at
a rate of 1 K min$^{-1}$, we evaluate Eq. (C1) every 6 s to obtain a temperature resolution of 0.1 K. We account for
the depletion of droplets using the following equation:

$$f_{i,\text{un}} = \frac{N_{i,\text{un}}}{N_0} = \exp(-V_{\text{d}} J_{\text{hom}} \Delta t) \, f_{i-1,\text{un}} \tag{C2}$$

where $f_{i,\text{un}}$ is the fraction of droplets that remained unfrozen at $T_i$, $f_{i-1,\text{un}}$ is the unfrozen fraction of droplets at
$T_{i-1}$, and $\Delta t = 6$ s.

For comparison with our experiments, we use the homogeneous nucleation rate parameterization by Ickes et al.

501 (2015):

$$J_{\text{hom}} = C \exp\left(-\frac{\Delta g^{\#}}{k_{\text{B}} T}\right) \exp\left(-\frac{\Delta G}{k_{\text{B}} T}\right) \tag{C3}$$


where $C = 10^{35}$ cm$^{-3}$ s$^{-1}$, $k_{\text{B}}$ is the Boltzmann constant, $T$ is temperature, and $\Delta g^{\#}$ and $\Delta G$ are the diffusional
activation energy and thermodynamic energy barrier, respectively, calculated as follows (Zobrist et al., 2007):

$$\Delta g^{\#} = \frac{892 \text{ K } k_{\text{B}} T^2}{(T - 118 \text{ K})^2} \tag{C4}$$

$$\Delta G = \frac{16\pi}{3} \frac{v_{\text{ice}}^2(T) \sigma_{\text{sl}}^3(T)}{\left(k_{\text{B}} T \ln S(T)\right)^2} \tag{C5}$$


where the molecular volume of ice $v_{\text{ice}}$ and the saturation ratio $S$ (ratio between the equilibrium vapour pressure
of supercooled liquid and that of ice) depend on temperature using the parameterizations outlined in Zobrist et al.
(2007), while the solid–liquid interfacial tension $\sigma_{\text{sl}}$ is calculated using the parameterization from Reinhardt and
Doye (2013):

$$\sigma_{\text{sl}} \, [\text{N} \cdot \text{cm}^{-1}] = 3 \times 10^{-6} - 1.8 \times 10^{-8}(273.15 - T) \tag{C6}$$




*Code and data availability*. Plot data are compiled in the ETH Research Collection data repository at
doi:10.3929/ethz-b-000545467. Python scripts are available upon request. *Note from authors: The link will be*
*activated after acceptance of the manuscript for final publication*.

*Author contributions*. FNI and NS are co-first authors of the manuscript and contributed equally to the instrument
design, generation of data, data analysis, and writing of the original draft; as such, they may each list their name
first in their CV. All authors contributed to project conceptualization, methodology, writing (review and editing),
and have approved the final version of the manuscript.

*Competing interests*. At least one of the (co-)authors is a member of the editorial board of Atmospheric
Measurement Techniques. The peer-review process was guided by an independent editor, and the authors have
also no other competing interests to declare.

*Acknowledgements*. We acknowledge work by Roland Walker who machined and 3D-printed portions of the
instrument, Fredy Mettler who provided support for the Peltier element, and Benedikt Waser who calculated the
calibration equations for the thermocouples. We also appreciate technical advice from Cyril Brunner and Kunfeng
Gao, as well as helpful discussions with Naama Reicher, Ulrich Krieger and Thomas Peter.

*Financial support*. NS acknowledges support from an ETH Postdoctoral Fellowship (20-1 FEL-46) and a Natural
Sciences and Engineering Research Council of Canada (NSERC) Postdoctoral Fellowship.



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
