# Peer review of "The Microfluidic Ice Nuclei Counter Zürich (MINCZ): A platform for homogeneous and heterogeneous ice nucleation"

_Atmospheric Measurement Techniques, 2022_

## Author Comment (AC1)

Authors' response to reviewers' comments for "The Microfluidic Ice Nuclei Counter Zürich (MINCZ): A platform for homogeneous and heterogeneous ice nucleation" by Florin N. Isenrich, Nadia Shardt, Michael Rösch, Julia Nette, Stavros Stavrakis, Claudia Marcolli, Zamin A. Kanji, Andrew J. deMello, and Ulrike Lohmann

We are grateful for Gabor Vali's comments and constructive suggestions that improved our manuscript. Below we outline our point-by-point replies and revisions to the manuscript. Page and line numbers refer to the uploaded document with tracked changes.

**Reviewer #1: Gabor Vali**

**Comment**

The instrument described in the paper is based on a good idea and it was built with care. The paper presents a thorough description in clear language appropriate for an AMT publication. The main novelty of this instrument is to separate the droplet production microfluidic device from the testing section where the cooling of the sample and the observation of freezing events takes place. The advantage derived is a better control of the sample temperature, minimizing internal temperature gradients that are the limiting factor to accuracy in some other droplet freezing devices.

On the production section, the choice of materials is crucial. This is well described in the paper but would find it helpful to clarify two things: Why is a surfactant (line 216) needed for a water in oil suspension? How are air bubble introduced (line 221) and why? In the end, are the water droplets in contact with the tubing and air, or also with some oil? How particle-free is the air? Is the surfactant likely to be covering the droplets in the test section?

**Authors' response**

A surfactant is needed to aid droplet formation and prevent the droplets from coalescing, especially at the outlet of the microfluidic device where the tubing is inserted. Surfactants are widely used (and needed) to stabilize the aqueous phase in microfluidic settings (e.g., Reicher et al. Atmos. Meas. Tech., 11(1), 233, 2018; Tarn et al. Micromachines, 12(2), 1, 2021). One alternative to the use of surfactants is to physically restrict droplet motion, as reported by Brubaker et al. (Aerosol Sci. Technol., 54(1), 79, 2019), but this physical restriction is not possible in the commercial PFA tubing that we use.

Regarding the air bubble, each syringe filled with support fluid (HFE or water) is pressed to first infuse the inlet PTFE tubing with the support fluid, and then the syringe plunger is withdrawn to take up a small volume of air. Each syringe plunger is withdrawn further to take up the primary fluid (either the surfactant–oil mixture or the aqueous sample). The air bubble only serves as a barrier between the support fluid and the primary fluid in the PTFE tubing. The air bubble remains in the inlet PTFE tubing and does not enter the microfluidic chip. In the end, the water droplets are in contact with the surfactant–oil continuous phase.

**Change to manuscript**

Page 8, lines 237–238: added "The air bubble remains in the inlet tubing and does not enter the microfluidic chip."

**Comment**

The precision of droplet sizes is indicated in Tables 1 and 2 in terms of the estimated variation in droplet diameters. The  $\pm 5 \,\mu\text{m}$  amounts to about 6.5%. This translates into a volume variation of about 20% which is not negligible in the evaluation of the results. This is a greater limitation to the overall performance of the instrument than is acknowledged in the paper. The authors' comment on this would he helpful.

**Authors' response**

We need to clarify that the  $\pm$  5 µm uncertainty that we report is a measurement uncertainty, instead of a physical variation in droplet diameter. This measurement uncertainty arises from the resolution of the CMOS camera and the magnification of the stereoscope, with an uncertainty in droplet radius of 2 pixels equating to our reported  $\pm$  5 µm in droplet diameter. To more precisely investigate the droplet size distribution, we have now observed the droplet sizes during production on an inverted bright field microscope (Ti-E, Nikon, Switzerland) equipped with a 20× 0.4 NA objective lens and a high-speed camera (Phantom Miro M310, Vision Research, USA). The standard deviation in one droplet population was 0.5 µm around the mean droplet diameter based on measurements obtained using ImageJ (Schneider et al. Nat. Methods, 9(7), 671, 2012), corresponding to a variation in droplet volume of 2%.

**Change to manuscript**

Page 11, lines 300–310: "The accuracy of mean diameter measurements is estimated to be  $\pm 5 \ \mu\text{m}$ . This measurement uncertainty arises from the resolution of the CMOS camera and the magnification of the stereoscope, with an uncertainty in droplet radius of 2 pixels equating to our reported  $\pm 5 \ \mu\text{m}$  in droplet diameter. However, the physical variability in droplet diameter for one droplet population is far less than this measurement accuracy. We independently monitored droplet generation on an inverted bright field microscope (Ti-E, Nikon, Switzerland) equipped with a 20× 0.4 NA objective lens and a high-speed camera (Phantom Miro M310, Vision Research, USA). We used flow rates of  $Q_{water} = 1.0 \ \mu\text{L} \ min^{-1}$ ,  $Q_{surfactant} = 1.5 \ \mu\text{L} \ min^{-1}$ , and  $Q_{spacer oil} = 2.0 \ \mu\text{L} \ min^{-1}$ , the same as those used for the water experiment on day 1 (Table 1). The standard deviation of droplet diameter in one droplet population was 0.5  $\mu\text{m}$  around the mean based on measurements obtained using ImageJ (Schneider et al., 2012), corresponding to a variation in droplet volume of 2%."

**Comment**

The small droplet size and the immersion of the tubing in a liquid are the main features regarding temperature accuracy. However, mention is made of a stack of glass slips (line 163) being placed below the tube. How does this limit the flow of the cooling liquid around the tube and to what extent does it introduce further temperature gradients. Could this be clarified?

**Authors' response**

The ethanol in the cooling bath does not actively flow around the tubing, but rather, heat is removed by the Peltier element located below the aluminium container. As the thickness of the glass slides placed at the bottom of the bath is uniform, we would not expect any horizontal temperature gradients where the tubing is placed (as confirmed by the fact that the freezing temperature is not affected by the location of the droplets in the array, as shown in the Appendix). However, regardless of the presence of glass slides, a vertical temperature gradient will develop within the bath upon cooling. Therefore, it is crucial to place the thermocouples in the same plane as the tubing (see Fig. 1c) to ensure

that the measured temperature is representative of the temperature of the droplets in the tubing. The position of the thermocouples in the same plane as the PFA tubing is ensured by the use of grooves in the PEEK holder that keep the thermocouples in place.

**Changes to manuscript**

Page 10, lines 266–267: added "During cooling of the ethanol bath, a vertical temperature gradient develops from the bottom to the top of the bath."

Page 10, lines 271–272: added "There are no horizontal temperature gradients, as confirmed by the fact that there is no spatial bias in freezing temperature (Appendix B)."

**Comment**

The spatial uniformity of temperature is demonstrated in Figs. B1 and B2. This display in terms of x and y coordinates` is somewhat unclear. Do both the x and y coordinates of all droplets in a sample are included? Probably yes. Also, is the x and y coordinate system given with respect to the internal dimension of the test chamber? A simple change to using the distance from the walls would be easier to understand.

**Authors' response**

We thank the reviewer for this suggestion, and we have therefore made the following changes to the manuscript.

**Changes to manuscript**

Pages 20–21: changed the *x*-axes of Figs. B1 and B2 to illustrate distance in millimeters instead of pixel coordinates. Additionally, we have reduced the size of the symbols to better discriminate between droplet locations.

Page 6: modified the schematic of the ethanol bath in Fig. 1b to include an outline of the field of view to help orient the reader.

**Comment**

The results and comparisons to other works are presented as the fraction frozen versus temperature. This is a straightforward manner of showing the results. However, it is specific to the volumes of the sample in the experiment. For even slightly polydisperse populations of drops the function looses generality and makes the calculation of the nucleation rate J for homogeneous freezing contain an error. It also influences the comparison of the three runs with microcline, as, according to Table 2, the drop volumes were about 20% larger for run 1 than for runs 2 and 3. The volume-dependence makes the FF(T) functions inadequate for comparisons with other experiments. It is not clear if any adjustments were made in Fig. 5 to overcome the problem.

In any case, this problem with the volume-dependence is not critical for this AMT paper. It would be more important for a science paper. To fully account for the volume variations in the samples is not a trivial matter. For the comparisons with literature results an appropriate caveat regarding the constant-volume assumption is probably sufficient. A more thorough step to bring results of different experiment on a comparable basis is conversion of the FF data into spectra (eq. 4 in Vali, G.: Revisiting the differential freezing nucleus spectra derived from drop-freezing experiments: methods of calculation, applications, and confidence limits, **Atmos. Meas. Tech.**, 12, 1219-1231, doi: 10.5194/amt-12-1219-2019, 2019.).

**Authors' response**

We agree that the effect of volume on the frozen fraction of droplets should be considered when frozen fractions are converted to nucleation rates. In a forthcoming publication, we show that the effect of small variations in volume is negligible for the homogeneous nucleation rate. For heterogeneous nucleation, we assume that the variability in the particle surface area per droplet most probably exceeds the aforementioned effect expected from variations in droplet volume. However, this effect could vary from one icenucleating particle type to another depending on its size distribution, and it needs to be assessed for the specific particle in question. Thus, we agree with the reviewer that the effect of volume variations warrants further investigation for heterogeneous nucleation rates, which we will consider in future work. As suggested, we add a short note for the reader to be aware of this effect when interpreting the reported frozen fractions.

**Change to manuscript**

Page 16, lines 462–464: added "We note that further interpretation of the frozen fraction and detailed theoretical analysis, such as calculation of particle surface area per droplet, may require considering the potential influence of variation in droplet volume, as outlined in, for example, Vali et al. (2019)."

**Comment**

Regarding the image analysis process. As described it is a demanding process. Is there some future improvement possible so that the apparatus would really become as user-friendly as it aims to be (line 134)?

**Authors' response**

For the present version of the apparatus, image analysis is indeed demanding, but future work is planned to tackle this problem. Because the difference in contrast between a liquid droplet and a frozen droplet depends on the polycrystallinity of the emerging ice phase, in very pure water, the ice phase has very few grain boundaries that scatter light. The contrast improves markedly when the water contains solutes (i.e., the droplet becomes much brighter). We aim to develop a fully automated image analysis algorithm based on the semi-automated approach described in this manuscript.

---

## Author Comment (AC2)

Authors' response to reviewers' comments for "The Microfluidic Ice Nuclei Counter Zürich (MINCZ): A platform for homogeneous and heterogeneous ice nucleation" by Florin N. Isenrich, Nadia Shardt, Michael Rösch, Julia Nette, Stavros Stavrakis, Claudia Marcolli, Zamin A. Kanji, Andrew J. deMello, and Ulrike Lohmann

We are grateful for the anonymous reviewer's comments and constructive suggestions that improved our manuscript. Below we outline our point-by-point replies and revisions to the manuscript. Page and line numbers refer to the uploaded document with tracked changes.

**Reviewer #2 Comment**

This paper describes the development of an improved microfluidic device that has lower temperature gradients and less water and gas permeability. The paper is suitable for AMT and should be publishable after some revisions.

This paper does need a thorough editing for content and especially length. The paper should be considerably shortened before final publication. There are also grammatical errors and statements that are not quantified. Some of these are called out specifically below but I encourage the authors to edit and shorten the paper before resubmission.

**Authors' response**

We have implemented the specific suggestions of the reviewer for shortening the paper, and we have checked the whole manuscript for opportunities to condense where possible.

**Comment**

**Abstract:**

1. The Abstract is too long and reads like an introduction to a paper. Please remove extraneous details and cut the text could by 50% which can be done without loss of important content. Please concentrate on the instrument being presented.

**Authors' response**

We have now significantly shortened the abstract; please see the tracked changes in the updated manuscript.

**Comment**

2. "requirements: (i) high accuracy and precision in measuring droplet temperatures within 0.2 K (ii) ability to reach the homogeneous freezing point of pure water, with a median freezing temperature of 237.3±0.1 K..." These appear to be capabilities, not requirements? Also, here and throughout the paper : the uncertainties here seem contradictory. Is the latter the uncertainty or spread in the freezing temperature? The numerous uncertainties, specifically in temperature, used throughout the paper need to be explicitly stated for clarity.

**Authors' response**

We are now more exact in our language around uncertainty, and we explicitly refer to accuracy and precision instead. Immediately below, we only list changes to the abstract, and later, in response to a follow-up comment by the reviewer we list the remaining changes.

**Changes to manuscript**

Page 1, lines 31–32: changed "high accuracy and precision in measuring droplet temperatures within 0.2 K" to "high accuracy of 0.2 K in measured droplet temperature"

Page 1, lines 33–34: changed "median freezing temperature of  $237.3 \pm 0.1$  K" to "median temperature of 237.3 K with a standard deviation of 0.1 K".

**Comment**

3. The Title and Abstract state MINCZ is going to provide homogeneous freezing data but the above quote and paper then says this is restricted to only pure water – something that doesn't exist in the atmosphere. This means a qualification – "…a platform for homogenous water and …" needs to be added.

**Authors' response**

There is consensus that the homogeneous nucleation rate of pure water is important in the atmosphere (e.g., Murray et al. Nature, 434(7030), 202, 2005). We demonstrate that MINCZ can be used to observe both homogeneous and heterogeneous freezing by investigating pure water and microcline suspensions. The platform is not limited to these specific cases and could also be used in the future to investigate the freezing behaviour of other solutions and suspensions.

**Comment**

4. "to detect mediocre and poor ice-nucleating particles" please define what mediocre and poor means (in temperature and saturation)? Please note that throughout the text words that are qualitative are often used when quantitation is necessary.

**Authors' response**

In shortening the abstract, we have deleted this phrase. However, we also used this wording in the main text of the manuscript, and we have now better defined what we intend to convey.

**Changes to manuscript**

Page 3, lines 78–79: changed "with mediocre or poor activity" to "that are active at temperatures between that of homogeneous freezing and the melting point of water"

Page 13, lines 376–377: changed "due to mediocre ice-nucleating particles" to "due to the presence of ice-nucleating particles"

Page 16, line 475: changed "due to mediocre or poor" to "catalysed by"

**Comment**

**Materials and Methods**

1. Many parts of Figure 1 seem extraneous and the figure overall confusing. Inclusions of things such as syringes, computer, cooling bath complicate the figure without adding any detail not in the text. Parts c and d are the most important and could be combined with a very simple composite of a and b to improve the figure. Please consider what really needs to be in the figure and can't be in the text and eliminate for clarity.

**Authors' response and change to manuscript**

We have considered the reviewer's suggestion, and we have revised Figure 1 to only display the essential components of the instrument. We have removed panel (a), and

retained panels (b) through (d), as suggested, and we have updated the caption and main text accordingly (see tracked changes in document).

**Comment**

Can the authors explain in the text why a ~75 μm droplet size was chosen? Is this a
fabrication limit? While it is understood microfluidic devices can't attain the small size of
most atmospheric droplets it is important to detail why this size was chose and what
implication this volume has in relation to the atmosphere.

**Authors' response**

The lowest possible droplet diameter can be seen as a balance between the available diameter of the PFA tubing and the practicalities of generating such small droplets. A diameter of 75  $\mu$ m was chosen, because it was one of the available dimensions of commercially available PFA tubing into which droplets are loaded after droplet generation. We also tested PFA tubing with an inner diameter of 50  $\mu$ m, but due to the high pressure drop arising from such a small inner diameter, stable droplet generation became more challenging. The high pressure also increased the frequency of PDMS delamination from the glass slide. Additionally, it is difficult to detect the freezing of smaller droplets unless a higher magnification objective is used (and then fewer droplets can be investigated simultaneously due to the smaller field of view).

From the perspective of homogeneous ice nucleation, the droplets themselves should be small enough to avoid heterogeneous freezing caused by impurities in the pure water (such as the gradual increase in frozen fraction at higher temperatures, as seen in Peckhaus et al. (Atmos. Chem. Phys., 16(18), 11477, 2016) and Brubaker et al. (Aerosol Sci. Technol., 54(1), 79, 2019)). At the same time, to investigate heterogeneous ice nucleation, it is better to investigate larger droplet volumes so that the surface area of ice-nucleating particles is distributed more uniformly amongst the generated droplets.

**Comment**

3. What information is detailed in Figure 2 that is not given in the text? It appears this figure simply repeats what the text says in a flow chart format.

**Authors' response and change to manuscript**

We have considered the reviewer's suggestion, and we have deleted Figure 2 from the manuscript.

**Comment**

4. The caption of Figure 3 is a method description, which belongs in the text, not a figure caption.

**Authors' response**

We have shortened the caption of Fig. 3 (now Fig. 2) by condensing two of the sentences into one, removing unnecessary explanation, and we have moved one sentence to the main text.

**Changes to manuscript**

Fig. 3 (now Fig. 2) caption: replaced the second and third sentences in the caption with "In the first step, locations where droplets potentially froze are automatically screened (highlighted in blue pixels for the two consecutive images and in green pixels for comparison to the image two time steps prior to  $I_t$ )."

Page 12, lines 340–341: deleted "To reduce the number of potential droplets that must be classified by the user"

Page 12, lines 349–351: added "Together, the above criteria aid in removing false positives from consideration and limit the number of potential freezing events that need to be presented to the user for visual classification."

**Comment**

Results

1. See also abstract. Can the authors explain the temperature uncertainties in the paper, specifically this section? For example, 2 significant figures in " $237.41 \pm 0.04$ " is in excess of the earlier statements on the equipment capabilities of .2. Figure 4 then appears to show a yet larger range in data which seems to be the most important uncertainty. A comprehensive explanation of the uncertainties and data ranges would greatly improve this paper.

**Authors' response**

We clarify that the accuracy of our reported temperature is 0.2 K. To better convey the meaning of the number following the  $\pm$ , we instead explain it in words.

**Changes to manuscript**

Page 1, lines 31–32: changed "high accuracy and precision in measuring droplet temperatures within 0.2 K" to "high accuracy of 0.2 K in measured droplet temperature"

Page 1, line 33–34: changed "median freezing temperature of  $237.3 \pm 0.1$  K" to "median temperature of 237.3 K with a standard deviation of 0.1 K".

Page 10, line 276: changed "uncertainty in our temperature measurement" to "accuracy of our temperature measurement"

Page 13, lines 360–361: changed "reproducible within a narrow temperature range of  $237.3 \pm 0.1$  K" to "237.3 K with a precision of 0.1 K (standard deviation of the three experiments)"

Page 13, lines 364–365: changed "an even narrower median temperature range of 237.41  $\pm$  0.04 K" to "a better precision of  $\pm$  0.04 K (standard deviation) in median temperature"

Page 14, lines 395–396: changed "244.6 K  $\pm$  0.7 K" to "244.6 K, with a spread of  $\pm$  0.7 K (standard deviation)"

Figs. 4 and 5 (now numbered 3 and 4): last sentence in each caption, replaced "uncertainty" with "accuracy"

**Comment**

2. The captions in Figures 4 and 5 are too long and include detail that is not a description of the figure. This text needs to be move to the main text.

**Authors' response**

In the caption of Fig. 5, we were able to shorten some of the text and move the details of the Peckhaus et al. (2016) data to the main text.

**Change to manuscript**

Fig. 5 caption: deleted "where 0.2 nL aqueous droplets with 0.05 wt% microcline suspension were printed onto a solid substrate and cooled at 1 K min-1."

Page 16, lines 449-450: added "in printed 0.2 nL droplets"

**Comment**

Conclusions

1. Consider not using "homogeneous" to describe the droplet size here as it is then used for the freezing mechanism; this is confusing.

**Authors' response**

We agree with the reviewer's suggestion.

**Change to manuscript**

Page 16, line 466: changed "homogeneously-sized" to "monodisperse"

---

## Author Response (AR2)

**Authors' response to reviewer's comments for "The Microfluidic Ice Nuclei Counter Zürich (MINCZ): A platform for homogeneous and heterogeneous ice nucleation"** by Florin N. Isenrich, Nadia Shardt, Michael Rösch, Julia Nette, Stavros Stavrakis, Claudia Marcolli, Zamin A. Kanji, Andrew J. deMello, and Ulrike Lohmann

We are grateful for Gabor Vali's follow-up comment and constructive suggestion to improve our manuscript. Below we outline our reply and revisions to the manuscript. Page and line numbers refer to the uploaded document with tracked changes.

**Reviewer #1: Gabor Vali**

**Comment**

As already indicated in my comment on the original submission, the data presentation in Fig 4 in terms of fraction frozen has disadvantages. This becomes most acute for the comparison with the results of Welti et al. (2019). The agreement found (for one of the particle sizes) must be considered coincidental in view of the large difference in INP concentration and in INP surface area. Single particles per drop in one case versus a suspension of powder in the other. This problem is mentioned in the text but not in the figure caption. Since for many people, graphs sum up the work this is worth correcting. Also, for what can be gained from the comparison it takes much space to describe and explain. The comparison is quite secondary to the description of the new instrument and to demonstrating its performance.

**Authors' response**

We agree on the importance of being careful when interpreting frozen fractions between instruments, and we have added a sentence to emphasize this in the caption.

**Change to manuscript**

Page 13, lines 386–387: added "We emphasize that the total particle surface area in each case must be considered when comparing frozen fractions."

To shorten the caption of Fig. 4, other minor changes are tracked in the attached document.